# Predicting the tensile properties of heat treated and non-heat treated LPBFed AlSi10Mg alloy using machine learning regression algorithms

Vijaykumar S. Jatti[1]*, A. Saiyathibrahim[2], Arvind Yadav[3], Murali Krishnan R.[4],
B. Jayaprakash[5], Sumit Kaushal[6], Vinaykumar S. Jatti[7], Ashwini V. Jatti[8], Savita V. Jatti[9],
Abhinav Kumar[10,11,12]*, Soumaya Gouadria[13], Ebenezer Bonyah[14]*

1 Department of Mechanical Engineering, School of Engineering and Applied Sciences, Bennett University, India, 2 University Centre for Research and Development, Chandigarh University, Mohali, Punjab, India, 3 Department of Electrical Engineering, GLA University Mathura, India, 4 Department of Mechanical Engineering, Karpagam Institute of Technology, Coimbatore, Tamil Nadu, India, 5 Department of Computer Science & IT, School of Sciences, JAIN (Deemed to be University), Bangalore, Karnataka, India, 6 Centre for Research Impact & Outcome, Chitkara University Institute of Engineering and Technology, Chitkara University, Rajpura, Punjab, India, 7 Department of Civil Engineering, Symbiosis Institute of Technology, Symbiosis International (Deemed University), Pune, India, 8 Department of Instrumentation Engineering, D Y Patil Institute of Technology, Savitribai Phule Pune University, Pune, India, 9 Department of Civil Engineering, D Y Patil College of Engineering, Savitribai Phule Pune University, Pune, India, 10 Department of Nuclear and Renewable Energy, Ural Federal University Named after the First President of Russia Boris Yeltsin, Ekaterinburg, Russia, 11 Department of Mechanical Engineering and Renewable Energy, Technical Engineering College, The Islamic University, Najaf, Iraq, 12 Department of Mechanical Engineering, Karpagam Academy of Higher Education, Coimbatore, India, 13 Department of Physics, College of Science, Princess Nourah bint Abdulrahman University,, Riyadh, Saudi Arabia, 14 Department of Mathematics Education, Akenten Appiah Menka University of Skills Training and Entrepreneurial Development, Kumasi, Ghana

* vijaykjatti@gmail.com (VSJ); drabhinav@ieee.org (AK), ebbonya@gmail.com (EB)

## Abstract

In this study, the ability of machine learning algorithms to predict tensile properties of both heat-treated and non-heat treated LPBFed AlSi10Mg alloy is investigated. The data was analyzed using various Machine Learning Regression (MLR) models such as Linear Regression (LR), Gaussian Process Regression (GPR), Random Forest Regression (RFR), and Decision Tree (DT). The AlSi10Mg alloys, heat-treated and non heat-treated, had different tensile characteristics. The tensile characteristics were forecasted using trained and evaluated MLR models. Because the performance of various MLR models has been verified by several performance indicators, such as Root Mean Square Error (RMSE), $R^2$ (coefficient of determination), Mean Square Error (MSE), and Mean Absolute Error (MAE). Moreover, scatter plots were made for checking the accuracy of the forecast. The GPR model demonstrated better prediction performance than the other three models, i.e., higher $R^2$ values and lower error values for the heat-treated samples. For predicting the UTS value of non-heat treated samples, the LR model performs very well with $R^2$ of 1.000. In that case, GPR has the better predictive performance for the other tensile features in non-heat treated samples. Summing up, it is obvious that GPR is well capable of predicting tensile

**Data availability statement:** All data are in the manuscript and/or Supporting Information files.

**Funding:** The author(s) received no specific funding for this work.

**Competing interests:** The authors have no relevant financial or non-financial interests to disclose.

**Abbreviations: UTS**, Ultimate Tensile Strength, **ARD**, Automatic Relevance Determination; **LPBF**, Laser Powder Bed Fusion; **MAM**, Metal Additive Manufacturing; **GPR**, Gaussian Process Regression; **MLR**, Machine Learning Regression; **RSME**, Root Mean Square Error; SLM, Selective Laser Melting; MSE, Mean Square Error; ASTM, American Society for Testing and Materials; MBE, Mean Bias Error; SLS, Selective Laser Sintering; MAPE, Mean Absolute Percentage Error; EBM, Electron Beam Melting; ISO, International Organization for Standardization; DED, Directed Energy Deposition; MAE, Mean Absolute Error; AM, Additive Manufacturing; 3D, 3 Dimensional; ML, Machine Learning; OLS, Ordinary Least Squares; UK, United Kingdom; ABF, adial Basis Function; RBF, Radial Basis Function; CAD, Computer Aided Design.

properties of AlSi10Mg alloy with high precision. This indicates how important GPR is to additive manufacturing to achieve great quality.

---

## 1. Introduction

Additive Manufacturing (AM), popularly called 3D printing, started with approximately thirty years of meaningful operations and enormous attention from the public and commercial circumferences and its capability to make a highly complicated form out of it by simply joining one piece at a time. The progress of scientific technology and the ability to control a large number of materials are in part responsible for the growth in additive manufacturing. It pushes for personalized items, faster product development, sustainability, cheaper manufacturing costs and lead times, and innovative business models [1]. AM encompasses three groundbreaking concepts: universality, practicality, and efficiency [2]. Thus, these new technological strides are gradually replacing the traditional production methods [3]. In the last decade, this method of operation has seen a quick spread, especially in the medical equipment and wearable sectors, to several industrial areas, including medical science, automobiles, and aviation, in which substantial development has been registered [4]. However, some drawbacks are also connected with it, for example, reduced productivity, inferior quality, or lack of information on the final mechanical characteristics [5]. The fundamental concept of additive manufacturing is the construction of three-dimensional geometries through the addition of materials, typically layer by layer. The subsequent features are shared by additive manufacturing processes: A deposition substance modified by the manipulation of points, lines, or regions to fabricate components, along with a computer capable of storing data, manipulating geometric features, and providing user guidance during the process [6]. Instead of using subtracting or shaping techniques, it is possible to fuse, melt, or bind material in the form of wire or powder to a 3D Computer Aided Design (CAD) file layer by layer with little human intervention [7]. Seven process groups may be used to categorize additive manufacturing according to the ISO/ASTM 52900 standard: directed energy deposition, binder jetting, powder bed fusion, material extrusion, vat polymerization, material jetting, and sheet lamination [8].

Currently, contemporary AM methods have the capability to produce a wide range of goods using diverse materials, including metals, polymers, ceramics, and composites [9]. Undoubtedly, Metal Additive Manufacturing (MAM) has demonstrated the most substantial influence in numerous industries. For example, MAM has proven to be effective in the medical field for producing various surgical titanium implants in recent times [10]. The aviation industry has seen substantial advancements in the heat exchanger assembly of the GE9X engine used by the Boeing 777. Previously consisting of 300 pieces, it has now been streamlined into a single component. This new component is not only 40% lighter but also 25% cheaper [11]. The first printed metal bridge in the world was just made by a construction company using MAM. The 10.5-meter-long bridge in De Wallen, Amsterdam, crosses the Oudezijds Achterburgwal canal and is constructed from 308LSi austenitic stainless steel [12]. PBF, which stands for Powder Bed Fusion, is a widely utilized manufacturing method that

is gaining popularity in various industries such as aerospace, medical, automotive, industrial, tooling, and consumer goods [13]. PBF technologies can be further categorized based on their specific melting mechanism. Electron Beam Melting (EBM) refers to devices that use electron beams as an alternative. DED, which stands for Directed Energy Deposition, is a widely used method for fixing and restoring material to existing parts. It can utilize either wire or powder as the feedstock [14]. Researchers have been captivated by LPBF as a technology in AM for metallic materials. The application of this 3D printing technology involves the use of a SLM machine. This technology offers significant advantages compared to traditional manufacturing methods. For instance, it is feasible to manufacture intricate-shaped components without the need for additional processing and with little limitations on geometry. The technology provides efficient material utilization, great adaptability, and reduced production time. Furthermore, it is possible to manufacture high-density components with various geometries in the same batch, while achieving near-net-shape [15–17]. A build plate is coated with a thin layer of metal powder using these processes, and then a portion of the item is melted or fused into the powder using a laser light source [18]. LPBF has gained increasing interest in recent years due to its ability to manufacture a wide range of metallic components, mostly made of aluminium, titanium, stainless steel, super alloys, nickel, and cobalt chromium alloys [19]. Certain commercial alloys have demonstrated successful printing, achieving complete density, net form, and a high-quality surface finish. Nevertheless, the occurrence of excessive porosity and/or solidification cracking is commonly found in several widely used alloys, hence impeding the wider utilization of LPBF for the production of engineering parts [20,21]. Aluminium alloys with a composition close to the eutectic point, containing approximately 12.6% silicon by weight, such as Al–10%Si–0.5%Mg (referred to as AlSi10Mg), are commonly used in LPBF [22]. These alloys are preferred because their near eutectic composition enables the additive manufacturing of components with complete volumetric density and eliminates the occurrence of solidification cracking [23]. The inclusion of 10 weight percent silicon improves the fluidity of the liquid and prevents solidification cracking during the LPBF process [24]. In addition, AlSi10Mg manufactured by LPBF has demonstrated significant strength and ductility characteristics. Specifically, it has yield strength of around 250–300 MPa, a tensile strength of no more than 400 MPa, and a maximum strain at failure of 7% [25,26]. The strength is ascribed to a refined sub-grain cellular structure that is created by LPBF processing [27,28]. Consequently, AlSi10Mg has become the focus of numerous studies aimed at gaining a deeper understanding of the LPBF and AM techniques [29].

Machine learning is a vital component of modern research, leveraging advanced technology to improve computer systems. ML employs algorithms and neural network models to enhance system performance. These algorithms independently build mathematical models using sample data, known as "training data," enabling them to make decisions without explicit programming. ML draws inspiration from the interactions of brain cells [30]. Currently, ML techniques are utilized to carry out regression, classification, clustering, or dimensionality reduction on large datasets [31]. Several sectors of the industry rely heavily on machine learning models, and scientists are finding practical uses for these models in the actual world. To examine the required output variables, a range of machine learning algorithms are utilized to compare input boundaries and output variables. This entails the utilization of many ML models. The regression method is a prevalent and firmly established methodology for making forecasts and predictions. Determine the correlation between independent and dependent variables using this method. The purpose of a regression model is to determine the associations between a large number of input factors and a large number of output variables [32]. At present, multiple ML algorithms have been employed to forecast the correlation between the mechanical and physical characteristics and the process parameters of additive manufacturing components. To determine the actual functional correlations between input and response variables, which could differentiate component qualities, machine learning regression is an appropriate approach [33]. Machine learning algorithms provide excellent data-driven methods for optimizing the LPBF process and enhancing the quality of parts. Effective execution of these algorithms needs the acquisition, analysis, and organization of data; however, they are essential [34]. The process parameters for selective laser melting of Inconel 718 are optimized by means of the random forest network model developed by Kappes et al. [35]. The formation of pores and keyholes is due to the orientation and positioning of the component with recycled powder. Support vector machines were used by

Aoyagi et al. [36] to generate a process map for predicting the process variables in the additive manufacturing of biomedical CoCr alloy components with low pore density. The research used regression models and artificial neural networks in order to reduce the number of trials needed for optimizing the process parameters. Moreover, this study mainly focused on the optimization of the additive manufacturing process of a certain material through employing machine learning techniques. Machine learning methods were utilized by Rankouhi et al. [37] to increase process efficiency associated with the multi-material additive manufacturing effort. The researchers also introduced a method to predict process parameters in Selective Laser Melting (SLM) of 316L-Cu multi-materials using ML in an effective way. To anticipate the properties of the component, a multivariate Gaussian process was developed to model the component. Laser power, scan velocity, and hatching space were used as precise values in the model. Many investigations using MLR algorithms have been conducted with the purpose of predicting the process variables as well as the qualities of components forged with LPBF. However, the authors are unaware of the potential application of MLR to forecast the influence of tensile properties of AlSI10Mg alloy produced by the LPBF technique.

The primary objective of the current research is to forecast different tensile characteristics of AlSI10Mg alloy samples produced by the LPBF technique, both with and without solution heat treatment. This is achieved by employing renowned MLR algorithms. The performance of each model is assessed by utilizing an array of reliability metrics, including root mean squared error, mean square error, and mean absolute error.

## 2. Materials and methods

The AlSi10Mg powder used in this study was produced by Sandvik Osprey, Neath, UK, using gas atomization. It is to be noted that the content studied in this article belongs to the field of metal processing, does not involve humans and animals. This article strictly follows the accepted principles of ethical and professional conduct.

The powder has a uniform spherical form and a mean grain size of around 45 μm. The composition of the AlSi10Mg powder employed in this experiment is shown in Table 1. The specimens used in the research were created using a Renishaw RenAM 500E Laser Powder Bed Fusion machine, that has an ideal laser beam strength of 500 W and a beam diameter of 80 μm. The LPBF operation was performed employing the pulsed laser mode, with each following layer having a scan direction rotated by 67°. The stripe width was 5 mm, and no preheating of the build platform was used. Table 2 presents the specific process variables used in developing the specimens, along with their corresponding designations. To reduce oxidation, the construction chamber was filled with highly pure (>99.999%) argon gas after being evacuated to

**Table 1. AlSi10Mg powder composition.**

| Element | Si | Fe | Cu | Mg | Sn | Ti | Ni | Zn | Mn | Pb | Al |
|---|---|---|---|---|---|---|---|---|---|---|---|
| Composition (wt%) | 10.8 | 0.55 | 0.05 | 0.35 | 0.05 | 0.15 | 0.05 | 0.10 | 0.45 | 0.05 | Balance |

**Table 2. LPBF processing parameters used and their respective volumetric energy density.**

| S. No. | Sample Designation | Laser Power (W) | Exposure Time (μs) | Hatch Distance (mm) | Layer Thickness (mm) | Volumetric Energy Density (J/mm³) |
|---|---|---|---|---|---|---|
| 1. | A | 350 | 50 | 0.08 | 0.06 | 60.76 |
| 2. | B | 350 | 45 | 0.09 | | 48.61 |
| 3. | C | 350 | 40 | 0.12 | | 32.41 |
| 4. | D | 450 | 50 | 0.08 | | 78.13 |
| 5. | E | 450 | 45 | 0.09 | | 62.50 |
| 6. | F | 450 | 40 | 0.12 | | 41.67 |
| 7. | G | 500 | 50 | 0.08 | | 86.81 |
| 8. | H | 500 | 45 | 0.09 | | 69.44 |
| 9. | I | 500 | 40 | 0.12 | | 46.30 |

a pressure of 900 mbar. Prior to activating the laser, the procedure was done three times in order to reduce the oxygen level to a number below 20 parts per million. Subsequently, argon was introduced inside the chamber at a pressure that exceeded the ambient pressure by roughly 20 mbar. Fig 1 a) depicts the sample produced employing the LPBF technique, with a diameter of 8 mm and a length of 65 mm. In addition, a group of specimens underwent solution treatment at a temperature of 530ºC for duration of 6 hours, and were subsequently quenched.

## 2.1. Calculation of volumetric energy density

The volumetric energy densities achieved on each specimen with various processes configurations are outlined in Table 2. The volumetric energy density (J/mm³) is determined using Eq (1) [38].

$$VED = \frac{P}{\frac{PD}{E} \times d \times t}$$

(1)

where, P is laser power in watt, E is exposure duration in μs, PD is point distance in mm, d is hatch spacing in mm, and t is layer thickness in mm.

## 2.2. Tensile testing

The procedure for performing a tensile test on an AlSi10Mg component involves subjecting the substrate to axial tension until it attains its maximum tensile strength and breaks. This test measures the material's mechanical properties, including the UTS, yield strength, and elongation at the time of failure. During the test, a controlled force is applied while the AlSi10Mg component is safely fastened at both ends. However, as the strain is increased, the specimen elongates until it finally fractures. Finally, the acquired data lends itself to the understanding of how the material responds to strain, which is an important tool to assess the applicability of the material for different types of applications with areas such as aviation and the car industry that can benefit from lightweight and high-strength materials. The ASTM E83-02 standard was followed to fabricate tensile specimens from printed bars (Fig 1b)). Tensile tests were conducted on heat-treated and untreated parts using computerized servo-hydraulic digital universal apparatus. The equipment was run at an extension rate of 0.5 mm/min, while maintaining the ambient temperature.

## 2.3. Machine learning regression algorithm

Regression is a fundamental technique in machine learning which is used to solve common regression problems that appear in diverse range of applications. The goal is to predict the best output with some unknown corresponding input features also called as the process parameters. Consider a training set,

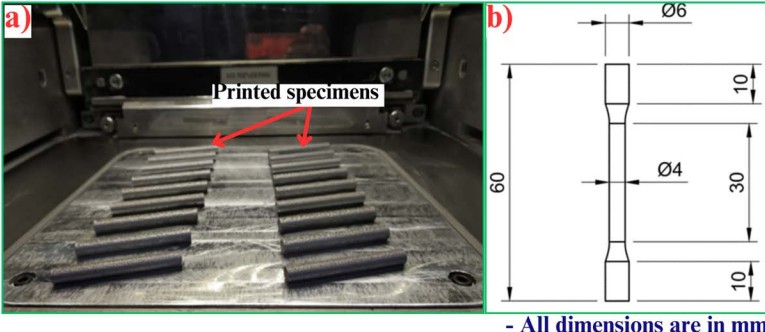

**Fig 1. a) Specimen printed using LPBF process, and b) Tensile test specimen.**

$$S = \{(x_1, \ y_1), \ \ldots, \ (x_n, \ y_n)\}$$
(2)

Where, N is the number of input features, $x \in R^S$ with corresponding output or target function $y_n \in R$ and for all $n = 1, 2, \ldots,$ N. Here, $y_n$ is the dependent variable/function and $x_n$ is the independent variable. Machine learning models are trained on the training dataset shown above in Eq (2). In the current investigation, the ultimate tensile strength of LPBFed tensile specimens is predicted using a variety of ML algorithms/models. These models provided prediction accuracy over 99% (R-Squared values of 0.99).

**2.3.1 Augmented data.** In practical applications, to train a model based on the training data which is already available makes it difficult and inefficient to provide a good output of results. So, to increase the efficiency of fitting/training a ML model, the process of data augmentation is carried out to generate a much greater number of lines of data based on the experimental data gathered by conducting experiments on the samples/workpieces. Data was augmented up to a limit of 100 iterations and then distributed in the ratio of 80:20 within training set and testing set respectively. Training and testing sets are used to train and test the model performance respectively. This helps in training a model efficiently without overfitting the training data, also giving good prediction accuracy and $R^2$ values. This also results in minimizing the losses (MSE. RMSE, MAE) in training and testing of a model.

Data augmentation is essential for enhancing the resilience and generalizability of machine learning models, especially in situations with scarce experimental data. Up to 100 iterations in data augmentation were used in this study to increase dataset variability while keeping the shape invariant of the original data distribution. The main problem in data augmentation is overfitting (that a model remembers the training data and never learns a more general pattern). Nevertheless, an auxiliary application of augmentation methods was able to reduce overfitting efficiently. This means that a primary reason to avoid bias from overfitting is the data augmentation itself that brings in more variability and therefore reduces the possibility that the model captures some specific patterns. Through controlled modifications of the dataset, the model faced a bigger set of reliable data points that helped it to generalize to unforeseen samples. The tactics designed for augmentation in this work were carefully designed to follow physical limits and actual material behavior. The process, however, ensured that generated data was authentic and representative, shrugging off any intentional distortions that could lead to wrong learning by the model.

Also, cross-validation techniques such as k-fold validation have been used to evaluate the effectiveness of the model for unseen data. The cross-validation requires methodical division of the dataset, ensuring that the model is tested against many subsets, but not at the expense of a single train-test division. For the purpose of evaluating model generalization, this method provides a more complete evaluation and reduces the risk of training data memory. In addition, regularization techniques were used where appropriate to prevent overfitting of very complicated models. After empirical tests, it was decided to limit data augmentation to 100 iterations so that the data expansion falls in a balanced line with computation efficiency. Generating hundreds of synthetic samples can increase redundancy and even lead to overfitting. Iterative testing showed that 100 iterations worked well because it improved the dataset enough while maintaining the performance of the model. The results indicate that within this regulated augmentation framework, the model maintained its predicted accuracy without overfitting, hence supporting the usefulness of the approach employed.

**2.3.2 Data pre-processing.** By lowering the three primary barriers to data interoperability such as uncertainty surrounding metadata, difficulties with data transfer, and missing data, data standards can foster synergies. Standardization of data semantics, properties, structure, formats, and/or interfaces may be necessary to maximize the benefits of data analysis, even while standards restrict private economic activity [39]. A dataset standardisation is commonly required for many machine learning estimators to ensure optimal performance, especially when the individual features do not closely approximate standard normally distributed data. Data standardization makes it easier to train a model based on a scaled dataset. In order to standardise the dataset, the mean is eliminated and the variance is scaled to one. The standard score of a model is given by Eq (3),

$$Z = \frac{(x - \mu)}{s} \tag{3}$$

where, $\mu$ is the mean of data, and s is the standard deviation.

**2.3.3 Machine learning metrics.** Performance metrics are used to evaluate a model and to learn how well it is trained. The performance metrics used in this study are,

**(i) Mean Absolute Error (MAE)**

It offers an absolute value of the average sum of errors, it is given by Eq (4),

$$MAE = \frac{1}{N} \sum_{i=0}^{n} (y_i - \ddot{y}_i) \tag{4}$$

**(ii) Mean Squared Error (MSE)**

The individual mean error term is squared and summed, given by Eq (5),

$$MSE = \frac{1}{N} \sum_{i=0}^{n} (y_i - \ddot{y}_i)^2 \tag{5}$$

**(iii) Root Mean Squared Error (RMSE)**

The root of mean squared error is given by RMSE in Eq (6),

$$RMSE = \sqrt{\frac{1}{N} \sum_{i=0}^{n} (y_i - \ddot{y}_i)^2} \tag{6}$$

**(iv) R² (R-Squared)**

R-squared is calculated by dividing the sum of squares of residuals ($SS_{res}$) by the total sum of squares of errors ($SS_{total}$) from the regression model and subtract the quotient by 1, as mentioned in Eq (7).

$$R^2 = 1 - \frac{SS_{res}}{SS_{total}} = 1 - \frac{\sum_{i=0}^{n} (y_i - \ddot{y}_i)^2}{\sum_{i=0}^{n} (y_i - \underline{y}_i)^2} \tag{7}$$

where, $y_i$ is the actual expected output, $\ddot{y}_i$ is the predicted output, $\underline{y}_i$ is the mean of total output, N is the total number of observations.

**(v) Mean Absolute Percentage Error (MAPE)**

$$MAPE = \frac{1}{n} \sum_{i=1}^{m} \left( \frac{y_i - \hat{y}_i}{y_i} \right) x\ 100 \tag{8}$$

Where, $y_i$ represents the actual tensile property values, $\hat{y}_i$ denotes the predicted values generated by the model, and $n$ signifies the total number of observations.

**(vi) Mean Bias Error (MBE)**

$$MBE = \frac{1}{n} \sum_{i=1}^{m} (y_i - \hat{y}_i) \tag{9}$$

Where, $y_i$ represents the actual tensile property values, $\hat{y}_i$ denotes the predicted values generated by the model, and $n$ signifies the total number of observations.

**(vii) Adjusted R²**

$$Adjusted\ R^2 = 1 - \frac{\sum_{i=1}^n (y_i - \hat{y}_i)^2}{(y_i - i)^2} \tag{10}$$

Where, $y_i$ represents the actual tensile property values, $\hat{y}_i$ denotes the predicted values generated by the model, $\ddot{y}_i$ denotes mean of actual values and $n$ signifies the total number of observations.

The coefficient of determination is an estimation that quantifies the extent to which the input variables/features meet the projected output variable. $R^2$ multiplied by 100 gives the prediction accuracy of the model. These performance parameters are defined and studied in the analysis of the different ML models used [40].

## 2.4. Linear regression algorithm

The goal of regression is to identify a function f that maps inputs $x \in R^d$ to values of the corresponding function, $f(x) \in R$. We consider that we have a collection of noisy observations shown and a set of training input features or process parameters $xn$. A potentially unmodeled process and measurement/observation noise are described by the random variable chosen as c (observation noises are considered as nil~0) in the present study. In the current study, LR is used to find the best function that not only models the training dataset, but also succeeds in generalizing the function to predict the best tensile strength values at unknown input process parameter features.

$$y_n = f(xn) + c \tag{11}$$

$$f(x) = 308.56 - 0.06 * y_1 - 0.052 * y_2 - 0.0018 * y_3 + 0.068 * y_4 + 0.278 * y_5 \tag{12}$$

Considering function $f(x)$:

$$f(x) = a_1 x_1 + a_2 x_2 + \ldots + a_i x_i,\ x \in R^d\ and\ for\ all\ i = 1,\ 2,\ \ldots n \tag{13}$$

The Eq (13) shows the linear regression function which was modelled using a LR model. Furthermore, the function was utilized to fit a curve based on the linear regression coefficients and the y-intercept, which the model provided for each input parameter [41].

The prediction model utilized a simple yet effective methodology, emphasizing interpretability and reducing the risk of overfitting. The model fundamentally depended on Ordinary Least Squares (OLS) as its principal optimization method. OLS, a traditional technique, proficiently identifies the optimal linear correlation between predictors and the target variable by minimizing the sum of squared residuals. While Ridge Regression (L2 regularization) was used to mitigate the effect of multicollinearity or overfitting on the dataset, which is generally common in datasets with multiple features. This regularization method adds a penalty term that shrinks the coefficients down and hence reduces the model complexity and improves the generalization performance. The combination suggests choosing to equilibrate the model fit with stability, opting for a parsimonious methodology. The utilization of these strategies indicates that the model is based on linear regression. The lack of references to the characteristics of a neural network, such as activation function and neuron count, indicates that a linear model is being used.

## 2.5. Gaussian processes regression algorithm

To establish the correlation among input features and target variables, the predictive model was based on Gaussian Process Regression (GPR). The RBF kernel with Automatic Relevance Determination (ARD) was the fundamental

component of the GPR model, as it included the option to change the significance of the different features. The kernel hyperparameters (the length scale and noise level σ²) were optimized by using the Limited-memory BFGS (L-BFGS) algorithm. The function's smoothness was regulated by the length scale lying in the range [0.1, 10], and the noise level relating to the intrinsic variability of data ranged in [10⁻³, 1]. The model was tested for strong performance and reduced overfitting by using a 5-fold cross-validation technique to understand the model's generalization ability on multiple data subsets. The optimization approach was run over 100 iterations for a balance between computing efficiency and convergence towards optimal values. This approach involved conducting hyperparameter tweaking and validation in a rigorous manner to produce a reliable and accurate predictive model.

A probabilistic approach towards ML is an efficient path to predict the output variable at certain input locations. Consider a data set D in Eq (14) of input features/variables independently distributed as a Gaussian distribution in Eq (15) given as,

$$D = \{x_1,\ x_2,\ \ldots x_m\},\ x\ \in R^m \tag{14}$$

$$p(x_i)\ = N(x_i,\ \mu,\ \sigma_2),\ \textit{where for all } i\ \in \{1,\ 2,\ \ldots m\} \tag{15}$$

The gaussian probability function is defined by the by some parameters like mean of the data (μ), sample variance ($\sigma^2$), these parameters are also known as the estimators of the gaussian distribution. A gaussian distribution resembles a normal distribution of the data which is a bell curve plotted on a 2D plot. This can be further displayed in terms of a contour plot and a surface plot. The gaussian distribution is given by Eq (16),

$$p_X(x) = \frac{1}{\sqrt{2\pi\sigma^2}}\exp\left(-\frac{1}{2}\frac{(x_i-\mu)^2}{\sigma^2})\right) \tag{16}$$

where, X is the set of random input features/variables and $x$ is the argument. The equation (15) describes a normal distribution of the data and is represented by the function equation (16).

A bi-variable (two-variable) gaussian surface plot (3D surface plot) is depicted in Fig 2 which shows a surface plot of the two input features 'Layer thickness' ($x_1$) and 'Hatch distance' ($x_2$). The 2D contour plot showing the correlation between the two variables $x_1$ and $x_2$ is depicted in Fig 2.

**2.5.1 Kernel.** Kernel also known as the covariance function is an essential component of this smoothing procedure, which captures our past understanding of the functions we are trying to represent. Comparable inputs should produce comparable outputs, which is what we want from regression predictions: they should be logical and smooth. Eq (17) represents the Radial Basis Function (RBF) kernel, sometimes referred to as the radial basis function, an extensively utilised parameter in Gaussian processes. The model does not exhibit infinite differentiability, resulting in smooth and differentiable fitting and predictions.

$$k(x,\ x') = \exp\left(\frac{-||x-x'||}{2l^2}\right) \tag{17}$$

The plot of the RBF kernel $k(x,\ x')$ is depicted in Fig 3 which represents a normal distribution of the RBF kernel plotted against x [42].

## 2.6. Decision tree algorithm

The training phase of the prediction model concentrated on maximizing a regression task utilizing a Mean Squared Error (MSE) criterion. The model's architecture, presumably a decision tree-based ensemble approach like Random

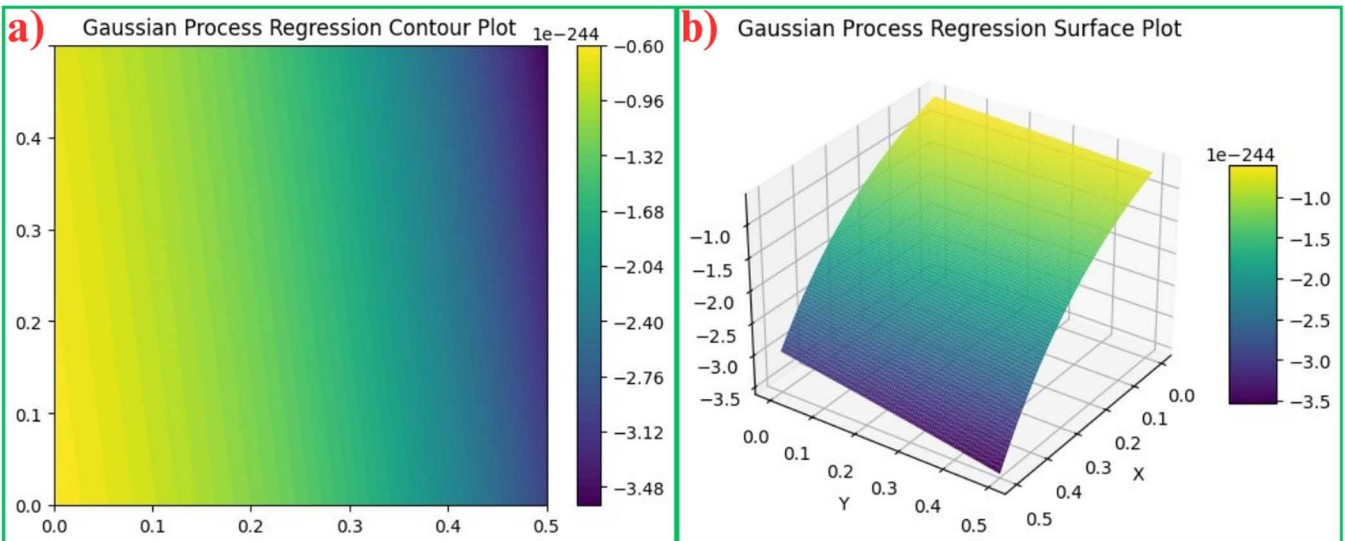

**Fig 2. Plots of Gaussian processes considering variables $x_1$ and $x_2$, a) 2-D contour plot of GP, and b) 3-D Surface plot of GP.**

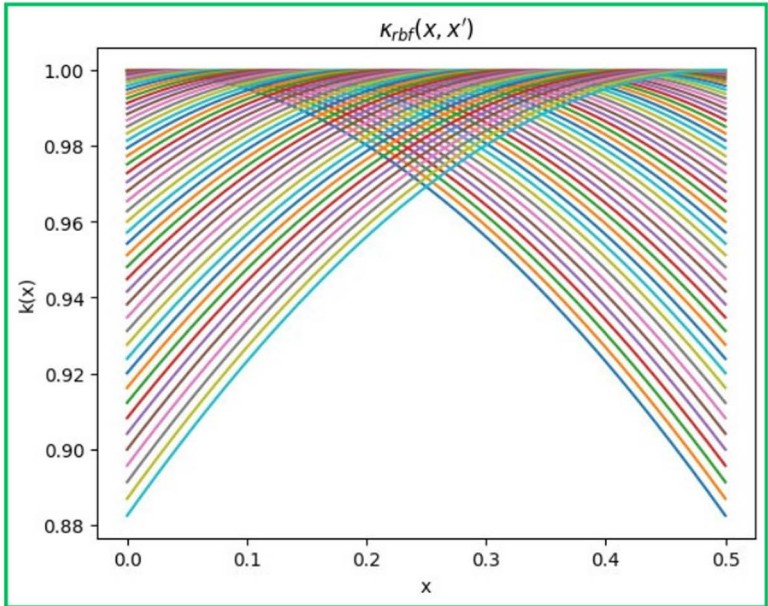

**Fig 3. Normal distribution of the RBF Kernel.**

Forest or Gradient Boosting, was examined by adjusting the maximum depth of individual trees from 3 to 20, hence regulating the model's complexity and capacity to discern complicated patterns. The minimum sample size necessary to divide an internal node was modified between 2 and 10, affecting the tree's granularity and its vulnerability to overfitting. The parameters of maximum depth and minimum samples per split directly influence the model's bias-variance trade-off; shallower trees and increased minimum samples encourage simpler models, potentially resulting

in higher bias, whereas deeper trees and reduced minimum samples pose a risk of overfitting to the training data. The Mean Squared Error (MSE) loss function, a conventional metric for regression tasks, directed the optimization process by measuring the average squared deviation between predicted and actual values. The dependence on depth and minimum samples indicates an emphasis on structural modifications of the decision trees to enhance prediction efficacy.

**2.6.1 Decision tree.** Consider a random training vector set $x_i \in R^n$, for all $i = 1, 2, \ldots m$ and target vector $y_i \in R^n$, now the decision tree partitions the input feature vector space such that the target vectors/variables corresponding to the same target value are grouped together. These partitions at each level of the tree are known as nodes. We consider each node as 'm' and the datapoint at each node as '$D_m$' with number of similar samples at each node as nm. For each data split 'β'; a threshold value '$t_m$' is added as a constraint and which consists of an input parameter 'k'. The data is partitioned into two sides as conventional lefts and rights. This can be depicted as,

$$\beta = (k, \ t_m) \tag{18}$$

$$D_m^{left}(\beta) = \{(x, \ y)|x_k \leq \ t_m \tag{19}$$

$$D_m^{right}(\beta) = \{(x, \ y)|x_k \geq \ t_m \tag{20}$$

Furthermore, the quality of the data split of the node is computed by the loss function given by,

$$L(D_m, \ \beta) \ = \ \frac{n_m^{left}}{n_m}H(D_m^{left}(\beta)) + \ \frac{n_m^{right}}{n_m}H(D_m^{right}(\beta)) \tag{21}$$

$$\beta' = argmin_\beta L(D_m, \ \beta) \tag{22}$$

Eq (21) is an argmin function that minimizes the loss L (Dm, β). The tree keeps on splitting recursively until the maximum allowable (max. depth) is reached and the tree is cut-off. A complete decision tree structure is depicted in Fig 4, which shows all the constraints for each feature and the number of splits till the squared error is minimized. A tree structure is visualized in the graphical plot (Fig 4).

**2.6.2 Decision tree regression.** The data splits for each feature (x) and the samples m per leaf/node n are visualized in Fig 5. It was noticed that the number of splits in the data is similar to the tree plot. At each node, the loss/squared error function gets reduced until the tree model is trained accordingly such as the squared error is minimized.

Similar to the contour plot (2D) and the surface plot (3D) visualized in the gaussian model 1.2, we plotted the 2D (Fig 5) and 3D (Fig 6) visualizations of a DT structure. It is a simple representation of the tree structure explaining the number of data splits at each node with a maximum depth of 3. Each plane in Fig 6 depicts the data split at each node with n number of samples consisting of two features plotted with respect to the UTS in each plot.

## 2.7. Random forest regression algorithm

The training method of the prediction model, although not comprehensively outlined, discloses essential architectural and operational decisions. The model utilizes an ensemble method, specifically implementing a Random Forest algorithm, as shown by the "Number of Trees" parameter, which varies from 100 to 500. This parameter indicates an emphasis on strong and generalizable predictions through the aggregation of numerous decision tree outputs. The "Maximum Depth"

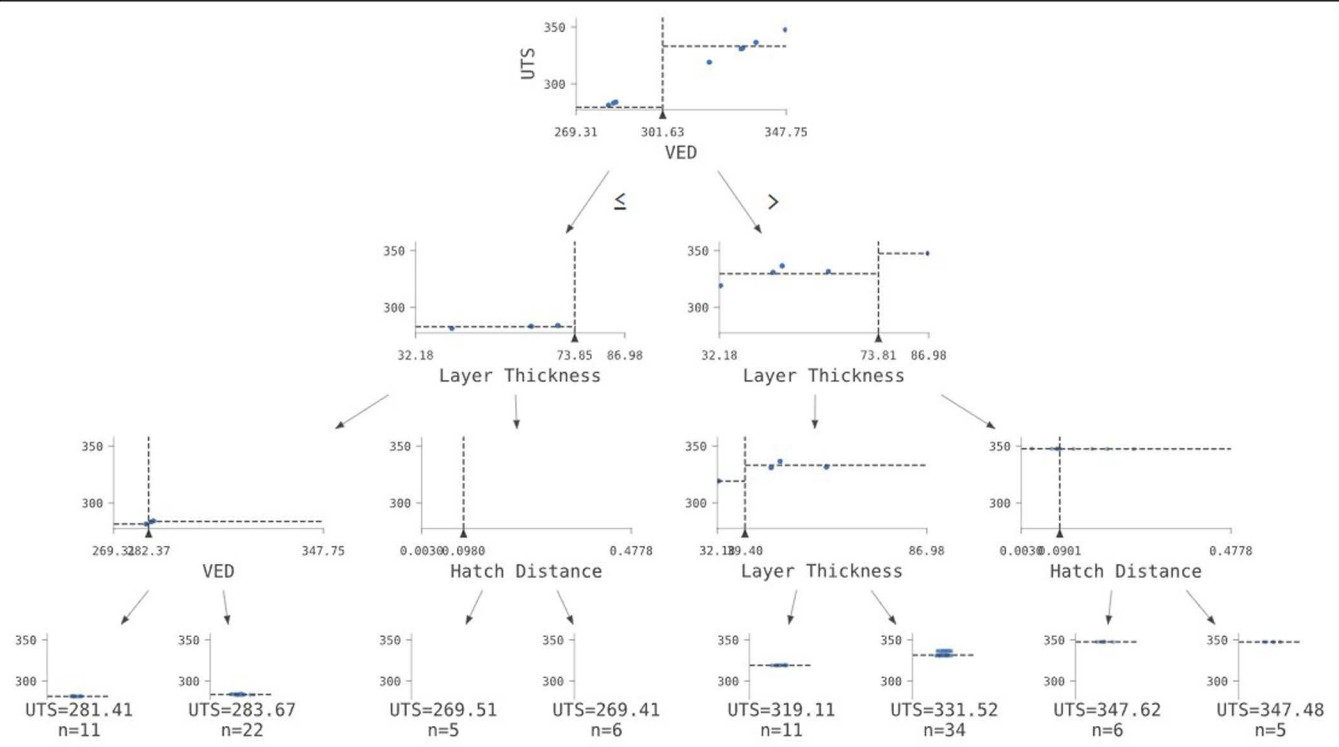

**Fig 4. Visualization of tree structure in graphical plot.**

range of 5–50 signifies an effort to equilibrate model complexity and the risk of overfitting, facilitating the identification of complicated associations in the data while reducing undue specialization. The "Minimum Samples per Split" range of 2–10 regulates the granularity of tree branching, affecting the model's responsiveness to nuanced patterns. "Feature Selection: Auto" denotes an automated methodology for identifying pertinent features, presumably employing approaches such as Gini significance or mean decrease impurity to optimize the model generation process. The "Validation: 80:20 Train-Test Split" denotes a conventional method for evaluating model performance, allocating 20% of the data for assessing generalization after training on the remaining 80%. This design, although devoid of specific information regarding the optimizer, activation functions, and learning rates (which are naturally less pertinent to random forests than to neural networks), exemplifies a realistic and established approach for constructing a predictive model.

**2.7.1. Feature importance.** Multiplying the probability of reaching a node by the weighted reduction in impurity at that node yields the feature relevance. The probability of a node may be calculated by dividing the total number of samples by the fraction of samples that reach the node. The value of a characteristic increases proportionally with its level of significance.

**2.7.2. Random Forest mathematical model.** For each decision tree in a random forest, each node's importance is calculated using Gini importance, considering only two child nodes as in a binary tree,

$$n_j = W_j H_j - W_{left(j)} H_{left(j)} - W_{right(j)} H_{right(j)} \tag{23}$$

where, $n_j$ is the importance value of node j, $W_j$ is the weight of number of samples at node j, $H_j$ is the loss or the impurity value at node j, j is the child node from left split on node j, and j is the child node from right split on node j.

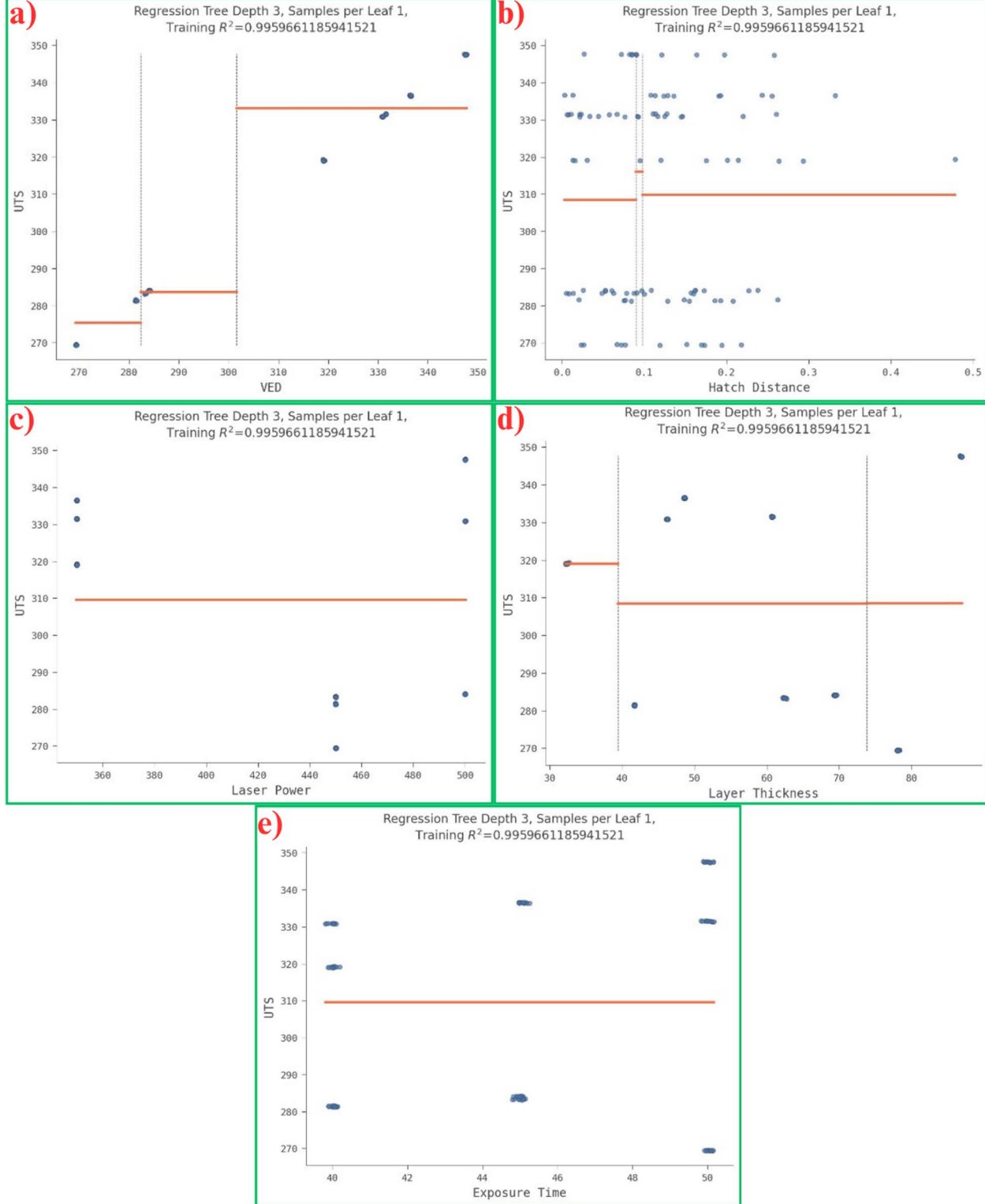

**Fig 5. 2D visualization of tree structure consisting of data splits with respect to each feature such as (a) VED vs UTS, (b) Hatch Distance vs UTS, (c) Laser power vs UTS, (d) Layer thickness vs UTS, and (e) Exposure time vs UTS.**

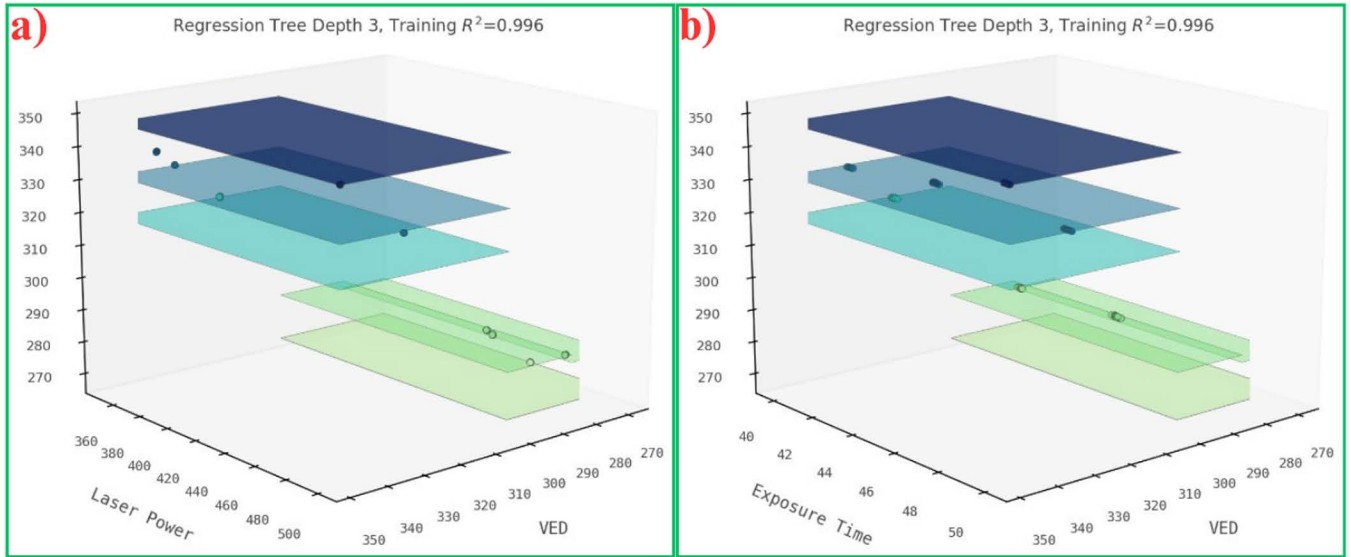

**Fig 6. 3D visualizations of tree structure considering 2 features such as (a) Laser power & VED, and (b) Exposure time & VED.**

The importance feature ($f_i$) of a decision tree is then calculated as,

$$f_i = \frac{\sum_{node\ j\ splits\ on\ feature\ i} n_j}{\sum_{k\ \in all\ nodes} n_k}$$

(24)

The workflow of random forest is similar to that in a decision tree. It has a very complex structure as there are lots of decision made based on the nodes, mean absolute error and the feature importance of each node [43].

## 3. Results and discussion

### 3.1. Experimental results

Table 3 presents the different tensile properties of specimens, both with and without solution heat treatment. This investigation investigates the ultimate tensile strength, yield strength and percentage of elongation for a variety of samples. The results of specimens that have undergone heat treatment and those that have not are compared. The ultimate tensile strength exhibits varied outcomes as a result of solution heat treatment. For certain cases, such as Sample A, the UTS increases from 331.54 N/mm² to 342.98 N/mm². This indicates that the heat treatment enhances the strength of the alloy. However, this is not applicable to every sample. Consider sample B as an illustrative example. The UTS decreases from 336.52 N/mm² to 322.24 N/mm² following the treatment. These findings indicate that heat treatment does not invariably enhance the strength of materials. Indeed, it can occasionally diminish their strength, contingent upon their initial state or the manner in which they are handled. The yield strength often improves with solution heat treatment. These patterns are observed in samples A, C, F, and H. The yield strength increases sequentially from 278.03 N/mm² to 302.39 N/mm², from 276.58 N/mm² to 294.88 N/mm², from 266 N/mm² to 295.50 N/mm², and from 244.05 N/mm² to 280.80 N/mm². The heat treatment enhances the alloy's ability to withstand plastic deformation, hence improving its performance under load. However, in certain instances, such as sample D and E, the young's modulus exhibits a decrease after undergoing heat treatment in comparison to its initial state. This occurrence could be attributed to variations in the heat treatment process or the inherent behavior of the material. The percentage of elongation, a measure of a material's ability to stretch, varies

**Table 3. Various tensile properties of heat treated and non-heat treated AlSi10Mg samples.**

| Sample Designation | With Solution Heat Treatment | | | Without Solution Heat Treatment | | |
|---|---|---|---|---|---|---|
| | Ultimate Tensile Strength (N/mm2) | Yield Strength (N/mm2) | Percentage of Elongation | Ultimate Tensile Strength (N/mm2) | Yield Strength (N/mm2) | Percentage of Elongation |
| A | 342.98 | 302.39 | 1.56 | 331.54 | 278.03 | 1.50 |
| B | 322.24 | 274.18 | 2.12 | 336.52 | 282.20 | 0.63 |
| C | 317.50 | 294.88 | 2.12 | 319.08 | 276.58 | 3.56 |
| D | 297.82 | 274.18 | 2.44 | 269.45 | 238.25 | 3.94 |
| E | 291.75 | 250.88 | 2.63 | 283.3 | 253.06 | 3.37 |
| F | 318.31 | 295.50 | 2.37 | 281.44 | 266 | 3.25 |
| G | 267.38 | 239.69 | 2.33 | 347.60 | 326.86 | 2.88 |
| H | 297.03 | 280.80 | 2.69 | 284.11 | 244.05 | 2 |
| I | 322.14 | 258.96 | 2.88 | 330.91 | 290.25 | 1.81 |

with heat treatment. Upon the application of heat, samples B and C undergo a chemical reaction. Sample B's elasticity increases from 0.63% to 2.12%, whilst C's decreases from 3.56% to 2.12%. This indicates that heat treatment, which is intended to enhance strength, might impact elasticity in many manners. For certain instances, such as sample B, the material becomes more malleable. For sample C, the observed rise in strength could indicate a decrease in flexibility. Heat treatment significantly influences the behavior of materials in intricate manners. Overall, the data indicates that solution heat treatment does not have a consistent impact on the characteristics of the AlSi10Mg alloy. It frequently enhances the strength of a material, particularly its yield strength, and occasionally its UTS. However, its impact on elongation can vary, resulting in either a rise or fall in ductility.

The inconsistencies found in the mechanical properties of the samples, both with and without heat treatment, can be ascribed to multiple sources. Initially, inconsistencies in the heat treatment process, including changes in soaking temperature, time, and cooling rate, can profoundly affect the microstructure and mechanical properties of the material. Inconsistent heat treatment can cause uneven dissolution of alloying elements, resulting in differences in tensile strength, yield strength, and elongation. Secondly, variances in material batches may account for these inconsistencies, as differences in raw material composition, impurity levels, and processing history might influence the ultimate mechanical performance. Other causes could also be related to certain aspects of the material microstructure, such as a grain size of the material used, the phase distribution, or residual stresses that may have been left during the production process. Furthermore, dimensional differences might produce a difference in the mechanical test results due to the discrepancies in the cross-sectional area of the sample, surface finish, or due to the stress concentrators in the sample. At times, the orientation of samples prior to testing or slight variation in strain rate, or the gripping mechanism, may enhance the disparity experienced. These variables collectively underscore the intricacy of material behavior and the necessity of meticulous process control to attain uniform mechanical characteristics [44].

Heat treatment is commonly utilized to improve the mechanical properties of materials, such as tensile strength and yield strength. However, its effect on ductility may be intricate, frequently resulting in negligible or even diminished enhancement, as illustrated in Table 3. Heat treatment primarily functions by dissolving secondary phases and homogenizing the microstructure, subsequently followed by rapid quenching to preserve a supersaturated solid solution. This procedure increases strength by refining grain structure and obstructing dislocation movement, although it may not boost ductility. Ductility often diminishes as a result of elevated brittleness induced by microstructural alterations, such precipitation hardening, residual stresses, or phase transitions. The results indicate that in the majority of sample designations, the percentage elongation, a measure of ductility either remains relatively constant or decreases during heat treatment, implying that the material attains greater strength but diminished capacity for plastic deformation. This constraint directly affects

practical applications, as a material exhibiting great strength but diminished ductility may be more susceptible to abrupt failure under impact or dynamic stress. Consequently, although heat treatment enhances strength, it must be wisely evaluated in instances where ductility and toughness are essential such as in aerospace and biomedical applications, where structural integrity and fracture resistance are essential considerations [45].

### 3.2. Prediction using various machine learning regression algorithms

The present work involved training and evaluating multiple machine learning algorithms to assess their efficacy in managing and interpreting the collected data. The goal is to ascertain the most appropriate algorithms to achieve the objectives of the study. This study involves assessing those algorithms for ML to gauge their effectiveness in governing and interpreting the acquired data. By assessing the effectiveness of the algorithms, the study can identify which one is most suitable for achieving the research targets and handling the unique features of the data. While assessing machine learning algorithms for forecasting the impact of volumetric energy density on tensile characteristics, it is important to analyze performance indicators which include RSME, MSE, $R^2$ and MAE. These indicators offer valuable perspectives into the models' ability to associate with and accurately forecast data from experiments. In order to assess the dependability of a ML framework specifically when it comes to forecasting outcomes such as the impact of volumetric energy density on tensile properties, the following metrics are essential: The models under consideration can be most effectively assessed using $R^2$, MAE, MSE, and RMSE.

**3.2.1. Ultimate tensile strength.** A study was undertaken in order to assess and contrast the efficacy of the machine learning regression models. This assessment specifically examined the prediction of the ultimate tensile strength of AlSi10Mg samples, both with and without heat treatment. Furthermore, it evaluates the disparity between the recorded and predicted UTS of the samples. Statistical measures such as $R^2$, RSME, and MSE were used for this comparison of both heat treated samples and non-heat treated samples, were indicated in the Tables 4 and 5.

Table 4 display the performance evaluation of regression models on UTS for heat-treated samples indicates that Gaussian Process Regression (GPR) outperforms alternate approaches. Its lowest RMSE (1.8069), MSE (6.8278), and Mean Absolute Percentage Error (MAPE) (43.7%), as well as the highest $R^2$ (0.9852) and Adjusted $R^2$ (0.9850), are all reported. Furthermore, GPR's predictive powers are validated by its small underestimate, as indicated by its Mean Bias Error (MBE) of -1.0173. Linear Regression (LR) is the second model, exhibiting a $R^2$ of 0.9325. However, its intermediate

**Table 4. Values of UTS prediction performance indicators for heat treated samples.**

| S.No. | Algorithms | Heat Treated Samples | | | | | | | |
|---|---|---|---|---|---|---|---|---|---|
| | | RMSE | MSE | R² | MAE | Accuracy | MAPE | MBE | Adjusted R² |
| 1. | Linear Regression | 4.7234 | 31.0534 | 0.9325 | 3.2923 | 93.0% | 69.70% | -1.4311 | 0.9318 |
| 2. | Gaussian Process Regression | 1.8069 | 6.8278 | 0.9852 | 0.7896 | 99.3% | 43.70% | -1.0173 | 0.9850 |
| 3. | Random Forest Regression | 7.9708 | 90.6216 | 0.8030 | 5.4453 | 99.9% | 68.32% | -2.5255 | 0.8010 |
| 4. | Decision Tree | 6.2098 | 48.1388 | 0.8953 | 3.6018 | 99.9% | 58.00% | -2.6080 | 0.8942 |

**Table 5. Values of UTS prediction performance indicators for non heat-treated samples.**

| S.No. | Algorithms | Non Heat Treated Samples | | | | | | | |
|---|---|---|---|---|---|---|---|---|---|
| | | RMSE | MSE | R² | MAE | Accuracy | MAPE | MBE | Adjusted R² |
| 1. | Linear Regression | 0.1621 | 0.0263 | 1.0000 | 0.1229 | 99.90% | 75.82% | -0.0392 | 1.0000 |
| 2. | Gaussian Process Regression | 0.5971 | 0.9504 | 0.9988 | 0.2944 | 99.00% | 49.30% | -0.3027 | 0.9988 |
| 3. | Random Forest Regression | 4.7435 | 29.9122 | 0.9609 | 3.9196 | 99.46% | 82.63% | -0.8239 | 0.9605 |
| 4. | Decision Tree | 2.6637 | 13.6209 | 0.9822 | 2.2987 | 99.60% | 86.30% | -0.3650 | 0.9820 |

accuracy is shown by its greater RMSE (4.7234) and MAPE (69.7%). The predictive efficacy of Decision Tree Regression (DTR) and Random Forest Regression (RFR) is inadequate, with RFR demonstrating the highest RMSE (7.9708) and MSE (90.6216), indicating possible overfitting and diminished generalization. Despite the seemingly high accuracy rates (99.9% for RFR and DTR), the increased MAPE values suggest that the forecasts lack stability. The exceptional success of GPR is due to its ability to model complex associations while adeptly handling uncertainty, whereas RFR and DTR may encounter overfitting due to their sensitivity to dataset fluctuations. The negative MBE values reported in all models show a propensity for underestimating; nevertheless, GPR mitigates this issue to the greatest extent. Generally, GPR is the most reliable model for forecasting the characteristics of heat-treated materials.

Table 5 shows the performance of various regression algorithms on UTS for non-heat-treated samples demonstrates that Linear Regression (LR) is the most successful model, providing an almost perfect fit with R²=1.0000 and the lowest RMSE (0.1621) and MSE (0.0263). This shows that the connection between input features and the target variable is nearly linear for non-heat-treated data, making LR the optimum choice. GPR also performed well, with R²=0.9988 and the lowest MAPE (49.3%), suggesting minimal relative error. However, its RMSE (0.5971) and MSE (0.9504) were greater than those of LR, making it marginally less successful. In contrast, Random Forest Regression (RFR) and Decision Tree Regression (DTR) demonstrated higher RMSE and MAPE values (82.63% and 86.30%, respectively), showing that they are less suited for predicting non-heat-treated samples. The inferior performance of RFR and DTR could be owing to their potential to overfit data, particularly if the underlying relations are more linear, making them less applicable than LR.

**3.2.2. Yield strength.** The Table 6 examines the effectiveness of different regression models on yield strength for heat-treated samples based on multiple parameters. GPR appears as the superior model with the maximum R² (0.9897), the smallest RMSE (1.2998) and MSE (3.8988), and a minimal MAPE (50.60%), demonstrating its excellent prediction accuracy with minimum error. LR and Random Forest Regression (RFR) indicate moderate performance, with R² values of 0.7128 and 0.7546, respectively, yet suffered from considerably higher RMSE (8.8124 for LR, 7.8540 for RFR) and MSE (109.1938 for LR, 93.2899 for RFR), resulting to increased prediction errors. Although RFR displaying an inflated accuracy (99.99%), its MAPE (75.29%) and MBE (-1.9407) imply potential overfitting. DTR scores the lowest, with R² (0.4012) and the greatest RMSE (14.3139) and MSE (227.6391), showing poor generalization and prediction reliability.

The outstanding performance of GPR can be credited to its capability to model complicated non-linear relationships with unpredictability quantification, whereas the weaker results of LR, RFR, and DTR could result from their constraints in capturing complex dependencies in the dataset, resulting to increased errors and reduced predictive reliability.

The performance of various regression models of non-heat-treated samples on yield strength is assessed using various metrics is depicted in Table 7, including RMSE, MSE, R², MAE, Accuracy, MAPE, MBE, and Adjusted R². Among the models, GPR surpasses all others, attaining its highest R² (0.9977) and Adjusted R² (0.9977), as well as the minimum RMSE (0.7897), MSE (1.4431), and MAE (0.3914), signifying its exceptional predictive accuracy and minimal error. Moreover, its low MAPE (49.56%) indicates a reduced percentage error relative to other models. RFR exhibits robust predictive capability with a R² of 0.8661; nevertheless, it possesses a greater error margin compared to GPR. LR performs moderately, with a R² of 0.8282, demonstrating acceptable accuracy but increased errors compared to GPR. Conversely, DTR has the

**Table 6. Values of yield strength prediction performance indicators of heat treated samples.**

| S.No. | Algorithms | Heat Treated Samples | | | | | | | |
|-------|-----------|------|-----|----|-----|----------|------|-----|-------------|
| | | RMSE | MSE | R² | MAE | Accuracy | MAPE | MBE | Adjusted R² |
| 1. | Linear Regression | 8.8124 | 109.1938 | 0.7128 | 7.3712 | 76.00% | 83.65% | -1.4412 | 0.7099 |
| 2. | Gaussian Process Regression | 1.2998 | 3.8988 | 0.9897 | 0.6577 | 98.14% | 50.60% | -0.6421 | 0.9896 |
| 3. | Random Forest Regression | 7.8540 | 93.2899 | 0.7546 | 5.9133 | 99.99% | 75.29% | -1.9407 | 0.7521 |
| 4. | Decision Tree | 14.3139 | 227.6391 | 0.4012 | 10.3816 | 95.53% | 72.53% | -3.9323 | 0.3951 |

**Table 7. Values of yield strength prediction performance indicators of non-heat treated samples.**

| S.No. | Algorithms | Non Heat Treated Samples | | | | | | | |
|---|---|---|---|---|---|---|---|---|---|
| | | RMSE | MSE | R² | MAE | Accuracy | MAPE | MBE | Adjusted R² |
| 1. | Linear Regression | 8.3904 | 105.9291 | 0.8282 | 5.6206 | 84.11% | 66.99% | -2.7698 | 0.8264 |
| 2. | Gaussian Process Regression | 0.7897 | 1.4431 | 0.9977 | 0.3914 | 99.70% | 49.56% | -0.3983 | 0.9977 |
| 3. | Random Forest Regression | 7.4895 | 82.5778 | 0.8661 | 4.7053 | 99.96% | 62.83% | -2.7842 | 0.8647 |
| 4. | Decision Tree | 11.4473 | 233.8796 | 0.6206 | 8.7530 | 99.14% | 76.46% | -2.6943 | 0.6167 |

least effective performance, evidenced by the lowest R² (0.6206) and the greatest MSE (233.8796) and MAE (8.7530), indicating inadequate predictive capability and elevated error variance. The exceptional efficacy of GPR arises from its capacity to model intricate, nonlinear relationships, whereas DTR often overfits or underfits the data, resulting in diminished accuracy.

**3.2.3. Percentage elongation.** Table 8 presents the performance of several regression models in evaluating the percentage elongation of heat-treated samples using multiple metrics. GPR was identified as the most effective model, attaining the highest R² (0.8903), the lowest RMSE (0.1382), and a comparatively low MAE (0.1029), signifying exceptional predictive accuracy. Moreover, its MAPE (74.46%) and Adjusted R² (0.8892) further validate its efficacy in reducing relative mistakes. RFR exhibited competitive performance with a R² of 0.7924; however, its RMSE (0.1842) and MAE (0.1375) were marginally elevated. Linear Regression had middling performance, achieving a R² of 0.8160, a comparatively low RMSE of 0.1748, and an accuracy of 72.00%, reflecting its adequate yet subpar predictive efficacy. Conversely, the DT model demonstrated the least effective performance, with the lowest R² (0.5312) and the greatest RMSE (0.2591), indicating inadequate generalization and elevated variance in predictions. The discrepancies in model performance can be ascribed to the dataset's complexity and each algorithm's capacity to capture non-linearity. GPR surpassed other methods owing to its probabilistic framework, which adeptly captures uncertainty and inherent patterns, whereas the Decision Tree's subpar performance can be attributed to overfitting and insufficient resilience in managing complex interactions.

From the compared models for estimating the percentage elongation of non-heat-treated samples in Table 9, GPR is the finest performing, having the highest R² of 0.9773 and the least RMSE of 0.5979, indicating splendid foresight and

**Table 8. Values of percentage elongation prediction performance indicators of heat treated samples.**

| S.No. | Algorithms | Heat Treated Samples | | | | | | | |
|---|---|---|---|---|---|---|---|---|---|
| | | RMSE | MSE | R² | MAE | Accuracy | MAPE | MBE | Adjusted R² |
| 1. | Linear Regression | 0.1748 | 0.0320 | 0.8160 | 0.1423 | 72.00% | 81.41% | -0.0325 | 0.8141 |
| 2. | Gaussian Process Regression | 0.1382 | 0.0191 | 0.8903 | 0.1029 | 91.13% | 74.46% | -0.0353 | 0.8892 |
| 3. | Random Forest Regression | 0.1842 | 0.0361 | 0.7924 | 0.1375 | 89.15% | 74.65% | -0.0467 | 0.7903 |
| 4. | Decision Tree | 0.2591 | 0.0816 | 0.5312 | 0.2001 | 93.29% | 77.23% | -0.0590 | 0.5264 |

**Table 9. Values of percentage elongation prediction performance indicators of non-heat treated samples.**

| S.No. | Algorithms | Non Heat Treated Samples | | | | | | | |
|---|---|---|---|---|---|---|---|---|---|
| | | RMSE | MSE | R² | MAE | Accuracy | MAPE | MBE | Adjusted R² |
| 1. | Linear Regression | 1.8264 | 4.3323 | 0.8054 | 1.4085 | 90.00% | 77.12% | -0.4179 | 0.8034 |
| 2. | Gaussian Process Regression | 0.5979 | 0.5064 | 0.9773 | 0.2546 | 95.74% | 42.58% | -0.3433 | 0.9771 |
| 3. | Random Forest Regression | 3.2691 | 14.3600 | 0.3550 | 1.3201 | 92.83% | 40.38% | -1.9490 | 0.3484 |
| 4. | Decision Tree | 4.1610 | 18.1627 | 0.1842 | 2.3307 | 93.17% | 56.01% | -1.8303 | 0.1759 |

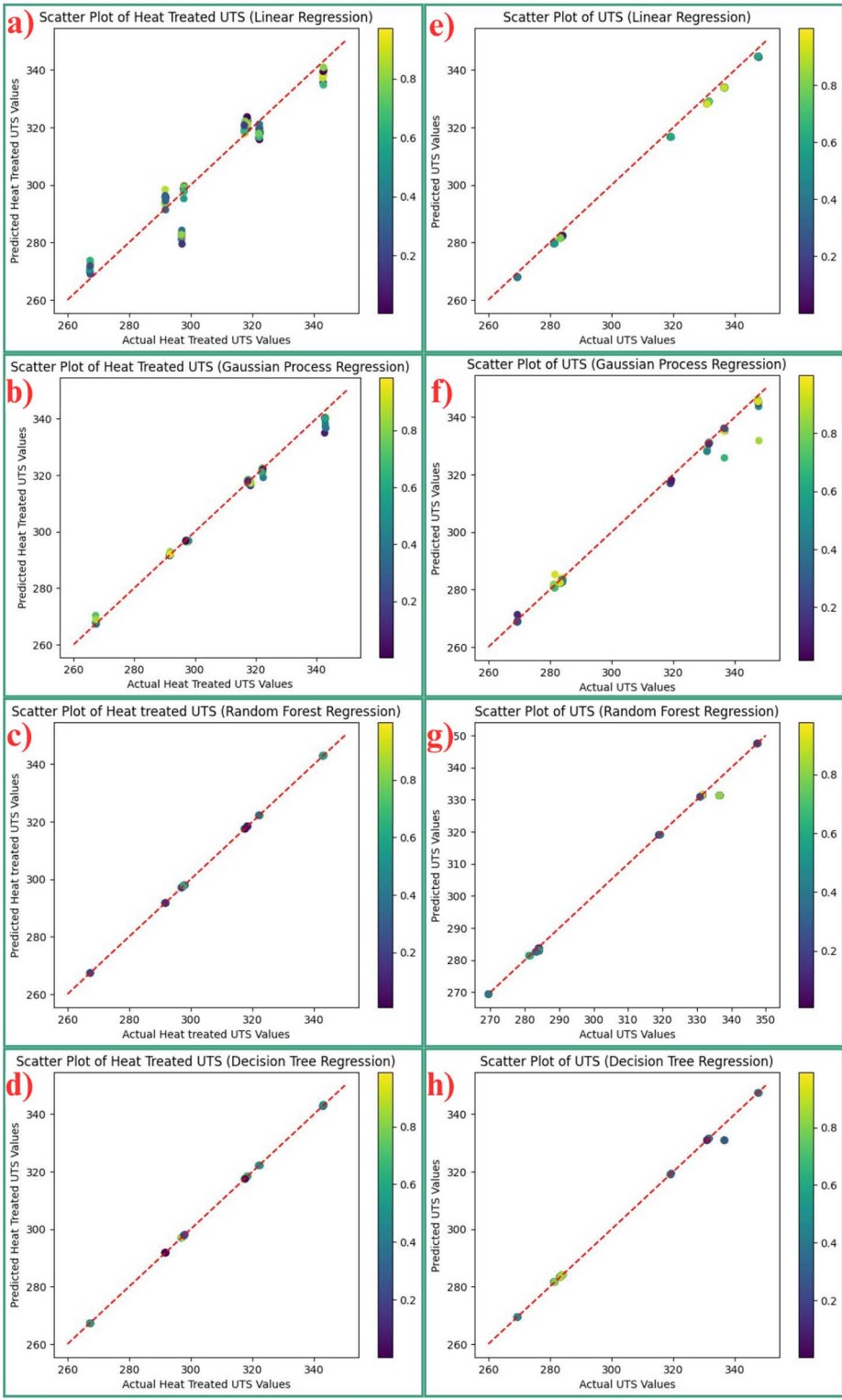

**Fig 7. Scatter plots for UTS prediction for heat treated and non-heat treated samples.** Scatter plots are used in MLR algorithms to illustrate the correlation between predicted and actual values. This helps in assessing the efficacy of the model. Scatter plots typically display the x-axis as the actual values and the y-axis as the predicted values. Optimally, the data points ought to align precisely along a 45-degree diagonal line, signifying precise predictions where the anticipated values correspond closely to the actual values. Deviations from the above approach differentiate between model forecasts

and actual data, emphasizing regions of excessive fitting or inadequate fitting. Scatter plots can be used to measure model accuracy, bias, variance, and overall goodness of fit, making them a crucial tool for evaluating and improving regression models [46]. Fig 7 illustrates the scatter diagrams produced by different MLR models used in this investigation to visually represent the relationship between the actual and predicted UTS of both heat-treated and non-heat-treated AlSi10Mg samples.

minimal inaccuracies. In addition, this model has a high overall accuracy of 95.74% and a comparatively low MAPE of 42.58%, which also confirms the stability of the results. Nevertheless, both RFR and DTR produce significantly worse predictive accuracy, as quantified by $R^2$ equal to 0.3550 for RFR and 0.1842 for DTR models and higher RMSE values, 3.2691 and 4.1610, respectively, which is why these models are vulnerable to overfitting and may not generalize well due to the data splitting approach. Although the LR model has a relatively good accuracy measure with an $R^2$ of 0.8054, it is outperformed by the GPR model equally in terms of all the critical performance parameters, indicating the ability of the GPR model in modeling the nonlinear relationship in the data set. The low accuracy attained by DT and RFR is a result of high variance and sensitivity to noise in data, while GPR compounds an acceptable level of accuracy with its probabilistic approach that embraces uncertainty and optimizes outcomes. Fig 9 depicts scatter plots created by MLR models used in this experiment to show the association between the measured and predicted percentage of elongation for both heat-treated and non-heat-treated AlSi10Mg samples. This figure unequivocally demonstrates that gaussian process regression surpasses all other models in accurately forecasting the percentage elongation of both heat treated and non-heat treated samples.

## 4. Conclusions

The study analyzed the use of machine learning algorithms to predict tensile characteristics of heat and non-heat treated AlSi10Mg alloy samples using laser powder bed fusion, using various regression models. As a result of this investigation, the following are the primary findings that were discovered.

- Heat treatment often improves yield strength; for example, Sample A rises from 278.03 N/mm² to 302.39 N/mm², because to the refinement of grain structure and the obstruction of dislocation movement. However, it exhibits variable effects on UTS, as demonstrated in Sample B, which declines from 336.52 N/mm² to 322.24 N/mm². Meanwhile, elongation diminishes, with Sample C decreasing from 3.56% to 2.12%.

- For predicting the UTS of AlSi10Mg specimens, demonstrating that GPR excelled for heat-treated samples, attaining a $R^2$ of 98.52% and an RMSE of 1.8069 N/mm². Conversely, LR demonstrated superior performance for non-heat-treated samples, with a $R^2$ of 100% and an RMSE of 0.1621 N/mm². The RFR and DT models demonstrated elevated RMSE and MAPE values, indicating possible overfitting and restricted generalization.

- In forecasting the yield strength of AlSi10Mg samples, GPR proved to be the most precise model, attaining a $R^2$ of 98.97% and an RMSE of 1.2998 N/mm² for heat-treated samples, while for non-heat-treated samples, it achieved a $R^2$ of 99.77% and an RMSE of 0.7897 N/mm². RFR and LR had modest performance, while DTR exhibited the least favorable performance, recording the lowest $R^2$ values (40.12% for heat-treated, 62.06% for non-heat-treated) and the greatest error margins.

- In predicting the percentage elongation of AlSi10Mg samples, GPR exhibited superior accuracy, attaining a $R^2$ of 89.03% and an RMSE of 0.1382 for heat-treated samples. For non-heat-treated samples, it achieved a $R^2$ of 97.73%, an RMSE of 0.5979, and an accuracy of 95.74%. RFR and LT demonstrated modest performance, whereas DTR underperformed, recording the lowest $R^2$ values (53.12% for heat-treated, 18.42% for non-heat-treated) and high RMSE values, signifying inadequate predictive dependability. The exceptional performance of GPR is due to its capacity to accurately represent non-linear connections and assess uncertainty.

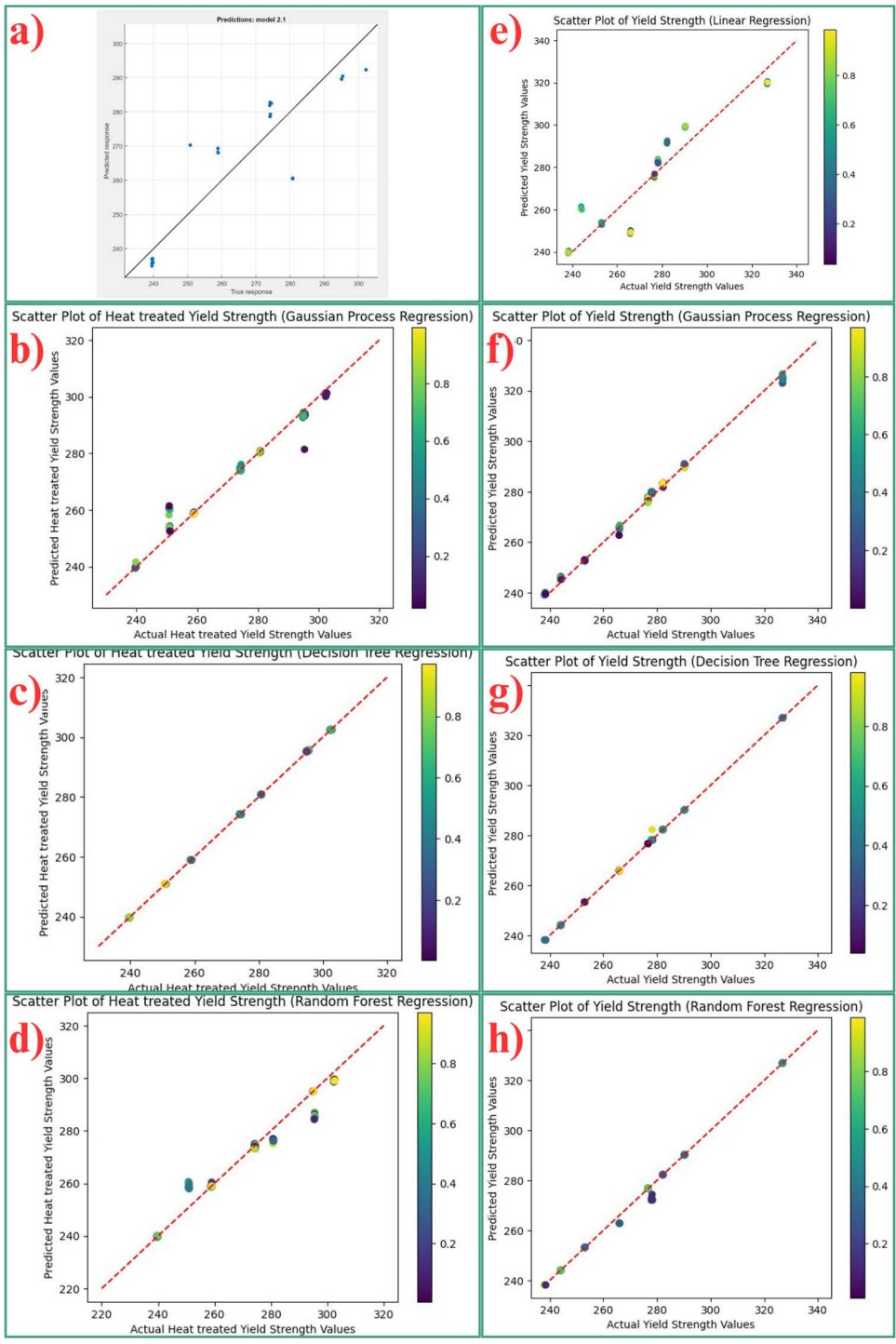

**Fig 8. Scatter plots for yield strength prediction for heat treated and non-heat treated samples.** The scatter plots generated by the machine learning regression models employed in this study to represent the correlation between the observed and estimated the yield strength of both heat treated and non-heat treated AlSi10Mg samples are illustrated in Fig 8. This plot illustrates the relationship of the projected values with the empirical data for each of the four models. Scatter plots demonstrate that non-linear models, such as gaussian process regression, random forest regression, and decision trees, outperform LR in terms of the accuracy of non-heat treated samples.

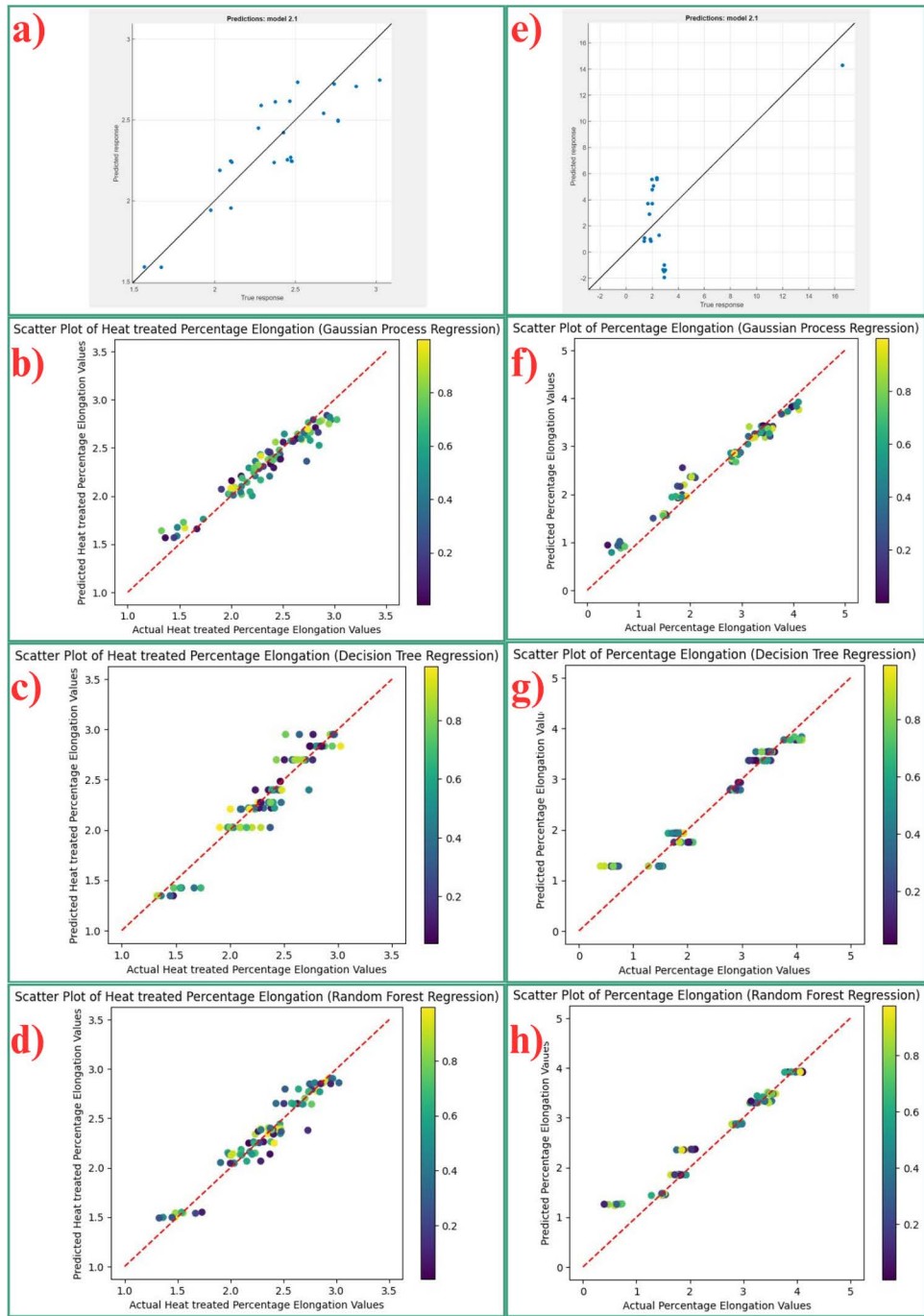

**Fig 9. Scatter plots for percentage elongation prediction for heat treated and non-heat treated samples.**

- The study found that GPR model is the most efficient machine learning model for predicting the tensile properties of heat-treated and non-heat-treated AlSi10Mg alloy, outperforming other models, highlighting the importance of selecting the right model.

- This study excludes other machine learning techniques like Support Vector Regression, XG Boost regression, Multi-Layer Perception Regression, and Hist Gradient Boosting Regression. Future expansion will include more models, benefiting experts in ML technology.

The results of this study have important pragmatic relevance for the design and production of LPBF AlSi10Mg alloy components. GPR is a reliable tool for improving processing settings and material properties, especially for samples that have been heated, because it can accurately predict tensile properties. Using these ML models will enable manufacturers to make data-driven decisions to raise mechanical performance, lower material waste, and increase manufacturing efficiency. In addition, linear regression is accurate for unheated samples, making it a quick way to check quality. These realizations can help engineers choose suitable post-processing treatments and design changes, improving the performance of components in aerospace, automotive, and other high-performance uses.

## Supporting information

**S1 Data. Numerical data used to generate all graphs and figures.**
(XLSX)

## Acknowledgments

Princess Nourah bint Abdulrahman University Researchers Supporting Project number (PNURSP2025R184), Princess Nourah bint Abdulrahman University, Riyadh, Saudi Arabia. The research funding from the Ministry of Science and Higher Education of the Russian Federation (Ural Federal University Program of Development within the Priority-2030 Program) is gratefully acknowledged.

## Author contributions

**Conceptualization:** A. Saiyathibrahim, Abhinav Kumar.

**Data curation:** Murali Krishnan R.

**Formal analysis:** Ashwini V Jatti, Savita V Jatti, Ebenezer Bonyah.

**Funding acquisition:** Arvind Yadav, Soumaya Gouadria.

**Investigation:** Vijaykumar S Jatti, Murali Krishnan R, Sumit Kaushal, Vinaykumar S Jatti, Ashwini V Jatti.

**Methodology:** Vijaykumar S Jatti, A. Saiyathibrahim, Abhinav Kumar.

**Resources:** Arvind Yadav, Jayaprakash B.

**Software:** Arvind Yadav, Jayaprakash B, Soumaya Gouadria, Ebenezer Bonyah.

**Supervision:** Vijaykumar S Jatti, Abhinav Kumar.

**Writing – original draft:** Vijaykumar S Jatti, A. Saiyathibrahim, Sumit Kaushal, Vinaykumar S Jatti, Savita V Jatti.

**Writing – review & editing:** Abhinav Kumar, Soumaya Gouadria, Ebenezer Bonyah.

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
