## [Decision Letter · Decision Letter 0]

20 Feb 2025

PONE-D-24-46880Predicting the Tensile Properties of Heat Treated and Non-Heat Treated LPBFed AlSi10Mg Alloy using Machine Learning Regression AlgorithmsPLOS ONE

Dear Dr. Bonyah,

Thank you for submitting your manuscript to PLOS ONE. After careful consideration, we feel that it has merit but requires minor revision to fully meet PLOS ONE’s publication criteria. Therefore, we invite you to submit a revised version of the manuscript that addresses the points raised during the review process.

We look forward to receiving your revised manuscript.

Kind regards,

Vasudev Vivekanand Nayak

Academic Editor

PLOS ONE

Journal Requirements:

“Princess Nourah bint Abdulrahman University Researchers Supporting Project number (PNURSP2025R184), Princess Nourah bint Abdulrahman University, Riyadh, Saudi Arabia.”

4. Please upload a new copy of Figure 4, 5a, 5b and 6 as the detail is not clear. Please follow the link for more information: 

https://blogs.plos.org/plos/2019/06/looking-good-tips-for-creating-your-plos-figures-graphics/

https://blogs.plos.org/plos/2019/06/looking-good-tips-for-creating-your-plos-figures-graphics/

Reviewers' comments:

Reviewer's Responses to Questions

**Comments to the Author**

1. Is the manuscript technically sound, and do the data support the conclusions?

Reviewer #1: Yes

Reviewer #2: Partly

2. Has the statistical analysis been performed appropriately and rigorously? 

Reviewer #1: Yes

Reviewer #2: Yes

3. Have the authors made all data underlying the findings in their manuscript fully available?

Reviewer #1: Yes

Reviewer #2: Yes

4. Is the manuscript presented in an intelligible fashion and written in standard English?

Reviewer #1: Yes

Reviewer #2: Yes

5. Review Comments to the Author

Reviewer #1: This study investigated the effectiveness of machine learning algorithms in predicting the tensile properties of heat-treated and non-heat-treated AlSi10Mg alloys, especially the ultimate tensile strength, yield strength and elongation. The alloy was produced by laser powder bed fusion ( LPBF ) technology. Multiple machine learning regression ( MLR ) models such as linear regression ( LR ), Gaussian process regression ( GPR ), random forest regression ( RFR ) and decision tree ( DT ) were used to analyze the data. The structure of the article is rigorous, and the results and discussions are rich. However, this manuscript still needs a lot of modifications :

1.The quality of the pictures in this article is generally poor, please upload high-quality pictures and add some picture labels.

2.The second chapter of the article has more conceptual content, please simplify the conceptual content and supplement the experimental data content involved in this article.

3.Although RMSE, MSE, MAE and R2 are used to evaluate the model, more statistical analysis methods can be added in the discussion to support the superiority statement of the model.

4.If the model has been hyper-parameterized, it can be clearly stated in the paper, which can help readers understand whether the reported performance indicators are the best results for each model.

5.The training process of the prediction model is less, such as : optimizer, activation function, neuron, number of iterations, learning rate and other data.

6.The paper mentions data augmentation, ' Data was augmented up to a limit of 100 Iterations... ', why there is no overfitting in the number of iterations within 100 times, please explain this part in more detail.

7.7.Although the effect of heat treatment on tensile properties ( such as tensile strength and yield strength ) was reported, the results showed inconsistencies, which may be confusing to readers. It is recommended to explain the reasons for these inconsistencies, such as changes in the heat treatment process, material batch differences, or factors such as the size / geometry of the sample. It is necessary to further explain why heat treatment does not generally improve the ductility and other properties, which will contribute to the practical application value of the research.

8.Although the conclusions summarize the research results well, the practical application implications of these results can be further discussed, such as : how the model can help guide the design of LPBF aluminum alloy components, and whether these findings can affect the decision-making in the manufacturing process.

Reviewer #2: The manuscript explores the application of machine learning techniques to model the mechanical properties of AlSi10Mg alloy samples fabricated via laser powder bed fusion. Various modeling approaches were employed, including linear regression, Gaussian process regression, random forest regression, and decision tree regression.

My considerations regarding the manuscript are as follows:

#1 Carefully review the text to identify and correct minor typographical errors, such as the one in the sentence following Equation 2.

#2 Enhance the quality and resolution of the figures, particularly Figures 4 and 5, to improve clarity.

#3 Improve the discussion on the tensile test results, emphasizing the differences between treated and untreated samples. Use new references to facilitate this discussion.

#4 The conclusions are too long, it is necessary to summarize them.

6. PLOS authors have the option to publish the peer review history of their article (what does this mean? ). If published, this will include your full peer review and any attached files.

**Do you want your identity to be public for this peer review?** For information about this choice, including consent withdrawal, please see our Privacy Policy .

Reviewer #1: **Yes: ** Jun GUO

Reviewer #2: No

---

## [Author Response · Author response to Decision Letter 1]

27 Mar 2025

Authors’ Response to Reviewers Comments

Manuscript ID PONE-D-24-46880R1

Manuscript Title Predicting the Tensile Properties of Heat Treated and Non-Heat Treated LPBFed AlSi10Mg Alloy using Machine Learning Regression Algorithms

Authors Vijaykumar S Jatti, Saiyathibrahim A, Murali Krishnan R, Vinaykumar S Jatti, Ashwini V Jatti, Savita V Jatti

The authors would like to thank the Editor and Reviewers for their valuable time to review the manuscript. The manuscript has been revised carefully based on the comments, and the details are as follows.

Responses to Comments

All the modifications are highlighted in the revised manuscript.

Reviewer#1 comments

Comment 1:

The quality of the pictures in this article is generally poor, please upload high-quality pictures and add some picture labels.

Authors’ Response:

The suggested comment is accepted.

As per the suggestion, all the figures are enhanced in high quality and also separately submitted with revised manuscript for further proceedings.

Kindly see page numbers 7, 13, 14, 16, 17, 18, 19, 20, 26, 29 and 32.

Comment 2:

The second chapter of the article has more conceptual content, please simplify the conceptual content and supplement the experimental data content involved in this article.

Authors’Response:

The suggested comment is accepted.

As per the advice, the content of entire chapter 2. Materials and Methods has been simplified and some experimental details are added in the revised manuscript.

Kindly see page numbers 9, 11, 12, 15, 20 and 21.

Comment 3:

Although RMSE, MSE, MAE and R2 are used to evaluate the model, more statistical analysis methods can be added in the discussion to support the superiority statement of the model.

Authors’ Response:

The suggested comment is accepted.

More statistical analysis methods such as Mean Bias Error (MBE), Mean Absolute Percentage Error (MAPE) and Adjusted R2 were included.

Kindly see page numbers 22, 23, 24, 25, 27, 28, 30 and 31.

Comment 4:

If the model has been hyper-parameterized, it can be clearly stated in the paper, which can help readers understand whether the reported performance indicators are the best results for each model.

Authors’ Response:

The suggested comment is accepted.

The Gaussian Process Regression (GPR) model using Bayesian Optimization/Grid Search, where key hyperparameters were tuned to maximize performance. The following hyperparameters were adjusted:

Kernel Function: Radial Basis Function (RBF) Kernel with automatic relevance determination

Length Scale: Tuned between [0.1, 10]

Noise Level (σ²): Optimized within [10⁻³, 1]

Optimizer: Limited-memory BFGS (L-BFGS) algorithm

Cross-validation: 5-fold cross-validation

This tuning process ensured that the reported performance metrics represent the best possible results for the model.

The following details are included in the revised manuscript. Kindly see page numbers 12 and 13.

Comment 5:

The training process of the prediction model is less, such as: optimizer, activation function, neuron, number of iterations, learning rate and other data

Authors’ Response:

The suggested comment is accepted.

The training process details for various models are included in the revised manuscript.

Kindly see page numbers 12, 13, 14, 15, 20 and 21.

Comment 6:

The paper mentions data augmentation, ' Data was augmented up to a limit of 100 Iterations... ', why there is no overfitting in the number of iterations within 100 times, please explain this part in more detail.

Authors’ Response:

The suggested comment is accepted.

Explanation related to data augmentation and overfitting prevention has been included in the revised manuscript.

Kindly see page number 9.

Comment 7:

Although the effect of heat treatment on tensile properties (such as tensile strength and yield strength) was reported, the results showed inconsistencies, which may be confusing to readers. It is recommended to explain the reasons for these inconsistencies, such as changes in the heat treatment process, material batch differences, or factors such as the size / geometry of the sample. It is necessary to further explain why heat treatment does not generally improve the ductility and other properties, which will contribute to the practical application value of the research.

Authors’ Response:

The suggested comment is accepted.

The reasons for the inconsistencies, such as changes in the heat treatment process, material batch differences, or factors such as the size/geometry of the sample, are also reasons for heat treatment not generally improving the ductility and other properties that are also discussed in the revised manuscript.

Kindly see page number 22 and 23.

Comment 8:

Although the conclusions summarize the research results well, the practical application implications of these results can be further discussed, such as: how the model can help guide the design of LPBF aluminum alloy components, and whether these findings can affect the decision-making in the manufacturing process.

Authors’ Response:

The suggested comment is accepted.

As per the advice, the practical application and suitability of the model can guide during the design of LPBF aluminum alloy components, and the effect of the study’s findings on the decision-making in the manufacturing process is discussed in the conclusion section.

Kindly see page number 34.

Reviewer#2 comments

Comment 1:

Carefully review the text to identify and correct minor typographical errors, such as the one in the sentence following Equation 2.

Authors’ Response:

The suggested comment is accepted.

Minor typographical error in the manuscript and sentence following Equation 2 was rectified.

Kindly see page number 8.

Comment 2:

Enhance the quality and resolution of the figures, particularly Figures 4 and 5, to improve clarity.

Authors’Response:

The suggested comment is accepted.

As per the advice, the quality and resolution of the figures 4 and 5 are improved. Kindly see page numbers 16, 17 and 18 of the revised manuscript.

Comment 3:

Improve the discussion on the tensile test results, emphasizing the differences between treated and untreated samples. Use new references to facilitate this discussion.

Authors’ Response:

The suggested comment is accepted.

The discussion on the tensile test results, emphasizing the differences between heat treated and non heat treated samples are included in the revised manuscript. Kindly see page numbers 22 and 23.

Further, new citations have been included for this discussion. Kindly see references 44 and 45 at page numbers 39 & 40.

Comment 4:

The conclusions are too long, it is necessary to summarize them.

Authors’ Response:

The suggested comment is accepted.

As per the suggestion, the conclusion section is summarized in the revised manuscript.

Kindly see page numbers 33 and 34.

Authors are thankful to the Editors and Reviewers.

---

## [Decision Letter · Decision Letter 1]

20 Apr 2025

Predicting the Tensile Properties of Heat Treated and Non-Heat Treated LPBFed AlSi10Mg Alloy using Machine Learning Regression Algorithms

PONE-D-24-46880R1

Dear Dr. Bonyah,

We’re pleased to inform you that your manuscript has been judged scientifically suitable for publication and will be formally accepted for publication once it meets all outstanding technical requirements.

Kind regards,

Vasudev Vivekanand Nayak

Academic Editor

PLOS ONE

Additional Editor Comments (optional):

Dear authors, please provide higher resolution versions of the figures used in this manuscript to the production office during the proofing stage.

Reviewers' comments:

Reviewer's Responses to Questions

**Comments to the Author**

1. If the authors have adequately addressed your comments raised in a previous round of review and you feel that this manuscript is now acceptable for publication, you may indicate that here to bypass the “Comments to the Author” section, enter your conflict of interest statement in the “Confidential to Editor” section, and submit your "Accept" recommendation.

Reviewer #1: All comments have been addressed

Reviewer #2: All comments have been addressed

2. Is the manuscript technically sound, and do the data support the conclusions?

Reviewer #1: Yes

Reviewer #2: Yes

3. Has the statistical analysis been performed appropriately and rigorously? 

Reviewer #1: N/A

Reviewer #2: Yes

4. Have the authors made all data underlying the findings in their manuscript fully available?

Reviewer #1: Yes

Reviewer #2: Yes

5. Is the manuscript presented in an intelligible fashion and written in standard English?

Reviewer #1: Yes

Reviewer #2: Yes

6. Review Comments to the Author

Reviewer #1: This study evaluated the capability of four machine learning regression models in predicting the tensile properties of heat-treated and non-heat-treated AlSi10Mg alloys. The Gaussian Process Regression (GPR) model demonstrated the highest prediction accuracy for heat-treated samples. For non-heat-treated specimens, the Linear Regression (LR) model showed superior fitting in predicting ultimate tensile strength, while GPR remained optimal for other properties. The research confirms the universality and high reliability of GPR in additive manufacturing material performance prediction, providing an efficient data-driven approach for process optimization. Upon review, the authors have addressed expert feedback through manuscript revisions and improvements. However, figure quality issues persist unresolved. Subject to these final amendments, this revised manuscript is recommended for acceptance.

Reviewer #2: Dear authors,

All the suggestions have been taken into account and have improved the understanding of the manuscript. Therefore, I recommend the manuscript for publication.

7. PLOS authors have the option to publish the peer review history of their article (what does this mean? ). If published, this will include your full peer review and any attached files.

**Do you want your identity to be public for this peer review?** For information about this choice, including consent withdrawal, please see our Privacy Policy .

Reviewer #1: No

Reviewer #2: No

---

## [Editor Report · Acceptance letter]

PONE-D-24-46880R1

PLOS ONE

Dear Dr. Bonyah,

I'm pleased to inform you that your manuscript has been deemed suitable for publication in PLOS ONE. Congratulations! Your manuscript is now being handed over to our production team.

Kind regards,

on behalf of

Dr. Vasudev Vivekanand Nayak

Academic Editor

PLOS ONE